# Molecular Characterization of *BRCA1* c.5339T>C Missense Mutation in DNA Damage Response of Triple-Negative Breast Cancer

**DOI:** 10.3390/cancers14102405

**Published:** 2022-05-13

**Authors:** Jeong Dong Lee, Won-Ji Ryu, Hyun Ju Han, Tae Yeong Kim, Min Hwan Kim, Joohyuk Sohn

**Affiliations:** 1Department of Human Biology and Genomics, Graduate School of Medical Science, Brain Korea 21 Project, Yonsei University College of Medicine, Seoul 03722, Korea; jdlee1@yuhs.ac; 2Avison Biomedical Research Center, Yonsei University College of Medicine, Seoul 03722, Korea; wjryu711@yuhs.ac (W.-J.R.); mghjhan@yuhs.ac (H.J.H.); tykim92@yuhs.ac (T.Y.K.); 3Division of Medical Oncology, Department of Internal Medicine, Yonsei University College of Medicine, Seoul 03722, Korea

**Keywords:** triple negative breast neoplasms, *BRCA1*, c.5339T>C, homologous recombination, RAD51

## Abstract

**Simple Summary:**

Recent studies re-classified the c.5339T->C; p.Leu1780Pro (L1780P) BRCA1 missense mutation as “likely pathogenic” in ACMG classification, which shows a high prevalence in the Korean population. This study aims to reveal the molecular mechanisms and therapeutic relevance of BRCA1 L1780P mutation in DNA damaging response of triple-negative breast cancer (TNBC).

**Abstract:**

BRCA1 L1780P BRCT domain mutation has been recognized as a pathogenic mutation in patients with breast cancer. However, the molecular significance of this mutation has not yet been studied in triple-negative breast cancer (TNBC) cells in vitro. We established MDA-MB 231, HCC1937, and HCC1395 TNBC cell lines expressing BRCA1 L1780P mutant. BRCA1 L1780P mutant TNBC cells showed increased migration and invasion capacity, as well as increased sensitivity to olaparib and carboplatin compared to BRCA1 wild-type cells. BRCA1 L1780P mutant TNBC cells showed decreased RAD51 expression and reduced nuclear RAD51 foci formation following carboplatin and olaparib treatment. The molecular interaction between p-ATM and BRCA1 was abrogated following introduction of BRCA1 L1780P mutant plasmid in TNBC cells, suggesting that the BRCA1 L1780P mutation disrupts the p-ATM-BRCA1 protein–protein interaction. We established an olaparib-resistant BRCA1 L1780P mutant TNBC cell line by chronic drug treatment. Olaparib-resistant cell lines showed upregulation of RAD51 expression upon olaparib treatment, and reduction in RAD51 expression in olaparib-resistant cells restored olaparib sensitivity. Collectively, these results suggest that the BRCA1 L1780P mutation impairs RAD51 recruitment by disrupting p-ATM-BRCA1 interaction, which is a crucial molecular factor in homologous recombination and olaparib sensitivity. Further therapeutic targeting of RAD51 in BRCA1 L1780P mutant breast cancer is warranted.

## 1. Introduction

Breast cancer is the most common malignancy among women worldwide [1]. Breast cancers are divided into several subtypes, including luminal A, luminal B, human epithelial growth factor receptor 2 (HER2) positive, and triple-negative breast cancer (TNBC) [2]. Each subtype has different histological and pathological features [3]. TNBC, which does not express the estrogen receptor (ER), progesterone receptor (PR), or HER2, shows particularly aggressive clinicopathological features [4]. Owing to the lack of ER, PR, and HER2, hormone- and HER2-directed therapies are ineffective against TNBC [5]. Recent studies have revealed that anti-PD-1/PD-L1 blockade improves patient survival in PD-L1-positive patients, and PARP inhibitors have also shown significant tumor response in patients with germline breast cancer associated gene 1/2 (BRCA1/2)-positive advanced TNBC. However, their activity is limited to patient subsets with PD-L1-negative or BRCA1/2-defined advanced TNBC, and the survival gain is limited by innate and acquired resistance. Therefore, it is important to identify novel therapeutic targets for TNBC treatment [6].

A previous study, 7 TNBC subtypes including basal-like 1 and 2 (BL1 and BL2), immunomodulatory (IM), mesenchymal (M), mesenchymal stem-like (MSL), and luminal androgen receptor (LAR) were characterized by differential gene expression profiles [7]. Several signaling pathways related to tumorigenesis of TNBC cells have been investigated [8], and impairment of the DNA repair pathways, which is mainly associated with BRCA 1/2, is now recognized as a major molecular target in TNBC [9]. Patients with germline BRCA1/2 mutations have an increased lifetime risk of breast cancer, as well as ovarian, pancreatic, and prostate cancer [10]. Furthermore, approximately 10% of patients with TNBC have BRCA1/2 mutation [11], which has the highest incidence among breast cancer subtypes [12]. Previous studies proposed synthetic lethality of BRCA1/2 mutant cancer cells upon PARP inhibition. The landmark trials OlympiaD and EMBRACA showed significant gain in progression-free survival following PARP inhibitor treatment compared to conventional chemotherapy, underscoring BRCA1/2 mutation as the foremost druggable target for TNBC [13].

In the initiation step of DNA double-strand break (DSB) repair process, MRE11, RAD50, and NBS1 form an MRE11-RAD50-NBS1 (MRN) complex to recognize the DSB site [14]. These DNA DSB sensors recruit and activate ATM for subsequent homologous recombination (HR) pathway [15]. Subsequently, after ATM activation, BRCA1 is phosphorylated by phospho-ATM, and the phosphorylated BRCA1 interacts with BRCA2 and partner and localizer of BRCA2 (PALB2) proteins [16]. The BRCA1-PALB2-BRCA2 protein complex then translocates to the nucleus and binds to RAD51, which is located in DNA-damaged foci in the nucleus [17,18] (Figure 1).

The RING and BRCA1 C-terminal (BRCT) domains in BRCA1 play a major role in the HR pathway [19]. BRCT domains act as phosphopeptide-binding modules following DNA damage. Therefore, ATM substrates with BRCT domains lead to the formation of protein complexes within DNA-damaged nuclear foci. As a result, mutations in the BRCT domain are involved in the impairment of BRCA1 [20,21]. Furthermore, impairment of BRCA1/2 function, known as “BRCAness”, affects the deficiency of the HR pathway, known as “HR deficiency” (HRD), by causing accumulation of genomic instability in cells, which is associated with tumorigenesis [22,23].

Because it is difficult to prove the relationship between BRCA1 variants and diseases, most variants are classified as variants of uncertain significance [24]. The American College of Medical Genetics has reclassified the BRCA1 L1780P missense mutation as a “likely pathogenic mutation” after analyzing clinical data. Recent studies have revealed that the BRCA1 c.5339T>C; p.Leu1780Pro (L1780P) missense mutation in the BRCT domain is closely associated with breast cancer in Korean patients [25,26]. These results demonstrate the high clinical relevance of BRCA1 L1780P missense mutation in Korean patients with breast cancer [27,28]. A translational study of the BRCA1 L1780P missense mutation is required to better understand breast cancer in Korean patients. This study aimed to investigate the mechanism of BRCA1 dysfunction due to the L1780P missense mutation or BRCA1 pathogenic mutation.

## 2. Methods

### 2.1. Cell Culture

TNBC cell lines, MDA-MB-231 (HTB-22), HCC1395 (CRL-2324), and HCC1937 (CRL-2336), with different BRCA1 statuses were purchased from the American Type Culture Collection (Rockville, MD, USA). Cell lines were grown in Roswell Park Memorial Institute 1640 medium (Thermo Fisher Scientific, Waltham, MA, USA) and Dulbecco’s modified Eagle’s medium (Thermo Fisher Scientific) supplemented with 10% fetal bovine serum (FBS) (Thermo Fisher Scientific) and 10,000 U/mL penicillin-streptomycin (Thermo Fisher Scientific). The cell lines were cultured at 37 °C in a 5% carbon dioxide (CO_2_) incubator. Sub-culturing was performed when the cell confluence reached approximately 85–90% in a 100 mm dish (Corning, New York, NY, USA). Cells were cryopreserved in cell freezing medium containing 90% FBS + 10% dimethyl sulfoxide (Duchefa Biochemie, Amsterdam, The Netherlands).

### 2.2. cDNA Transfection

All TNBC cell lines were transfected with a pCMV6-entry plasmid containing BRCA1 wild-type gene or BRCA1 L1780P missense mutant gene. The plasmids were purchased from OriGene (Rockville, MD, USA). Each parental cell line was seeded in a 6-well plate (Corning) at a density of 3–5 × 10^5^ cells/well. When the cells reached 70–80% confluence, 1–2 µg of plasmid DNA was transfected into the cells using Lipofectamine LTX (Thermo Fisher Scientific) according to the manufacturer’s protocol. After 8 h, the medium was replaced with medium containing geneticin (G418, 0.4–1 mg/mL) (Thermo Fisher Scientific). G418 selected the plasmid-transfected cells, which formed the cell clones. Stable cell lines with a low passage number were cryopreserved and used in the experiments.

### 2.3. Cell Migration Assay (Wound Closure Assay)

Parental and stable cells were seeded in six-well plates (Corning) at a density of 3–5 × 10^5^ cells/well. When cells reached 95–100% confluence in each well, a cutting tip was used to scratch the bottom of the well. Wound closure images were captured using a light microscope, eclipse TS-100 (Nikon, Tokyo, Japan), after 3 h, 6 h, 9 h, and 12 h. Image J software [29] (National Institutes of Health, Bethesda, MD, USA) was used to quantify the wound closure area.

### 2.4. Cell Invasion Assay

Cell invasion assays were performed using 24-well plates (Corning). Matrigel (Thermo Fisher Scientific) was added to the upper chamber. Parental and stably transfected cells were seeded in the upper chamber at a density of 1 × 10^5^ cells/well, and the bottom chamber was filled with plain or conditioned medium. After incubating for 24 h at 37 °C in 5% CO_2_, the cells in the bottom of the plate were stained with 0.1% crystal violet (Sigma Aldrich, St. Louis, MO, USA) and observed under a light microscope (Nikon).

### 2.5. Cell Proliferation Assay

Parental and stably transfected cells were seeded in a 96-well plate (Corning) at a density of 3 × 10^3^ cells/well. The growth rate of cells with different BRCA1 statuses were compared at 24–72 h using the CellTiter 96^®^ AQueous One Solution (Promega, Madison, WI, USA). Cell proliferation assay was performed according to the manufacturer’s protocol.

### 2.6. Quantitative Reverse Transcription Polymerase Chain Reaction (qRT-PCR)

Parental and stably transfected cells were washed with ice-cold phosphate-buffered saline (PBS). RNA was extracted using TRIzol^®^ reagent (Thermo Fisher Scientific) according to the manufacturer’s protocol. The RNA samples were dried and eluted in 10–30 µL of diethyl pyrocarbonate (DEPC)-treated water (Thermo Fisher Scientific). All preparative and handling procedures were performed under RNase-free conditions. The RNA concentration was measured using a Nanodrop™ 2000/2000c spectrophotometer (Thermo Fisher Scientific), and the total RNA samples were stored at −70 °C.

cDNA was synthesized using reverse transcription master mix (TaKaRa, Koyto, Japan; RR036A). PCR cycle, consisting of 37 °C for 15 min and 85 °C for 5 s, was performed using an Applied Biosystems Veriti^®^ 96-Well Thermal Cycler (Thermo Fisher Scientific). SsoAdvanced™ Universal SYBR^®^ Green Supermix (Bio-Rad, Hercules, CA, USA) was used for RT-PCR, and the level of RAD51 was compared following treatment with various drugs. GAPDH served as a control. PCR cycles consisted of 95 °C for 3 min, followed by 40 cycles of 95 °C for 10 s and 60 °C for 30 s and were performed using a CFX96 Touch Real-Time PCR System (Bio-Rad). Following primers were used to qRT-PCR: GAPDH, Forward—CGACCACTTTGTCAAGCTCA; Reverse—AGGGGAGATTCAGTGTGGTG and RAD51, Forward—AGCTTTCAGCCAGGCAAAT; Reverse—GCTTCAGCTTCAGGAAGACA.

### 2.7. Western Blot

Parental and stably transfected cells were washed with ice-cold PBS and lysed using cell extraction buffer (Thermo Fisher Scientific) containing a protease inhibitor cocktail (Roche, Basel, Switzerland). Protein concentration in the cell and tissue lysates were measured using the BCA Protein Assay Kit (Thermo Fisher Scientific). Then, 5× sodium dodecyl sulfate (SDS) sample buffer was added to the lysates and the samples were boiled at 100 °C for 5 min. Proteins were separated by sodium dodecyl sulfate-polyacrylamide gel electrophoresis (SDS-PAGE) and transferred to a nitrocellulose membrane (GE Healthcare Life Sciences, Pittsburgh, PA, USA) for 3 h. The membrane was incubated at 25–30 °C in 5% skim milk in 0.1% Tris-buffered saline with Tween 20 (TBS-T). The membranes were incubated with the primary antibody, which was diluted in 5% skim milk in 0.1% TBS-T, at 4 °C overnight. Then, the membranes were incubated with the secondary antibodies that were diluted in 5% skim milk in 0.1% TBS-T for 1 h at room temperature. Using primary and secondary antibodies are listed in Appendix A.

### 2.8. Co-Immunoprecipitation (Co-IP)

Co-IP was performed using the co-IP kit (Thermo Fisher Scientific) according to the manufacturer’s protocol. Parental and stably transfected cells were seeded in 6-well plates (Corning) at a density of 3–5 × 10^5^ cells/well. After treatment with the drug for 72 h, the cells were rinsed with PBS and lysed with IP lysis/wash buffer (0.025 M Tris, 0.15 M NaCl, 0.001 M EDTA, 1% NP-40, and 5% glycerol; pH 7.4). The primary antibodies were conjugated to AminoLink Plus Coupling Resin (settled resin supplied as a 50% slurry) for 2 h at room temperature and used to pull down the target proteins in the cell lysates. The antibody-coupled resin and cell lysates were incubated with gentle mixing overnight at 4 °C, after which Elution Buffer (pH 2.8, containing primary amine) was used to isolate the proteins. The isolated proteins were then subjected to SDS-PAGE.

### 2.9. Immunofluorescence (IF)

Parental and stably transfected cells were seeded in 8-well Lab-Tek chamber slides (Thermo Fisher Scientific) at a density of 4 × 10^4^ cells/well. After treatment with the drug for 72 h, the cells were rinsed with PBS and 4% paraformaldehyde was used to fix the cells. Cells were incubated with blocking solution for 30 min at room temperature, followed by incubation with the primary antibodies diluted in blocking solution, for 1 h at room temperature.

Then, the cells were incubated with the secondary antibodies, diluted in the blocking solution, for 45 min at room temperature. Cells were rinsed with PBS three times and the wells of Lab-Tek chamber slides were separated. A mounting medium containing DAPI (Vector Laboratories, Burlingame, CA, USA) was used to stain the nuclei. Using primary and secondary antibodies are listed in Appendix A.

### 2.10. Small Interfering RNA (siRNA)

siRNA transfection was performed using RNAiMAX Transfection Reagent (Thermo Fisher Scientific) according to the manufacturer’s protocol. RAD51 siRNA oligomers (RAD51_1 and RAD51_2) were used for knockdown of RAD51 expression. The samples were analyzed 48 h after transfection. For the cell proliferation assay, RAD51 knockdown cells were seeded in a 96 well plate (Corning) at a density of 3 × 10^4^ cells/well. After 24 h of incubation, serial concentrations of olaparib (0–10 μM) were used to evaluate drug sensitivity.

### 2.11. Olparib Resistant Cell Lines

Cells stably transfected with BRCA1 L1780P missense mutant were treated with olaparib for 6–12 months. Survival of clones and clonal expansion were observed. The MTS assay was used to determine whether the cell lines were resistant to olaparib, and the viability of olaparib-resistant and parental cells was compared. Olaparib-resistant cells were thus successfully established.

### 2.12. Statistical Analysis

Statistical analyses were performed using Microsoft Excel or GraphPad Prism 7 software (GraphPad Software, San Diego, CA, USA). All data are represented as the mean ± standard deviation (SD). All statistical analyses were two-sided, and statistical significance was set as *p* < 0.05.

## 3. Results

### 3.1. Stable Expression of Mutant BRCA1 Proteins in TNBC Cells and Its Impact on Cell Proliferation

We established stable BRCA1-expressing TNBC cell lines to understand the molecular effects of the BRCA1 L1780P mutation. FLAG-tagged wild-type and BRCA1 L1780P mutant plasmids were transfected into MDA-MB-231, HCC1937, and HCC1395 TNBC cells, which have different endogenous BRCA1 mutation status (Figure 2A and Appendix A): MDA-MB-231, BRCA1 wild-type; HCC1937, BRCA1 c.5382 insC; HCC1395, BRCA1 c.5251C>T, and BRCA2 c.4777G>T. Exogenous BRCA1 expression was confirmed by Western blotting with anti-FLAG antibody (Figure 2B and Appendix A). Sanger sequencing confirmed the successful introduction of mutant-BRCA1 in the stable cell lines (Figure 2C,D). These data indicated that stable BRCA1 wild-type and L1780P mutant cell lines were successfully established from TNBC cells. Next, we compared the cell proliferation capacity of each BRCA1-expressing cell line. BRCA1 L1780P mutant cells showed low cell proliferation capacity compared to BRCA1 wild-type MDA-MB-231 and HCC1937 cells. However, HCC1395 and each BRCA1-expressing cell line showed different cell proliferation capacity. (Figure 2E,F and Appendix A). These data suggested that different BRCA1 status affected cell proliferation capacity.

### 3.2. Cell Migration and Invasion Capacity of BRCA1 L1780P TNBC Cells

We next performed cell migration and invasion assays to evaluate the effect of BRCA1 L1780P mutation on the aggressive potential of TNBC cells. BRCA1 L1780P mutant cells showed higher migration capacity than BRCA1 wild-type cells (Figure 3A,B) in the wound closure assay. The cell invasion assay also revealed that BRCA1 L1780P mutant cells had higher invasion capacity than BRCA1 wild-type cells. These results indicate that BRCA1 L1780P mutant cells have high invasive potential despite their low proliferative capacity (*p* < 0.05) (Figure 3C,D).

### 3.3. Differential Effect of BRCA1 Mutation Status on Olaparib Sensitivity

We compared the viability of parental cells and each BRCA1-expressing cell lines after olaparib and carboplatin treatment (Appendix A). BRCA1 L1780P mutant cells showed higher sensitivity to olaparib than BRCA1 wild-type cells (Figure 4A,B and Appendix A), suggesting that the BRCA1 L1780P mutation renders the cells vulnerable to PARP inhibition similar to BRCA1 nonsense mutations. These data suggest that the BRCA1 L1780P mutation affects the sensitivity of cells to olaparib.

### 3.4. Effect of BRCA1 L1780P Mutation in RAD51 Induction upon Olaparib and Carboplatin Treatment

We next investigated the molecular effects of the BRCA1 L1780P mutation on BRCA1 and BRCA2-related DDR proteins. The expression of phosphorylated RAD51 and ATM was upregulated following olaparib and carboplatin (platinum-based) treatment in BRCA1 wild-type cells. Of note, the BRCA1 L1780P mutant expression significantly abrogated induction of RAD51 expression following carboplatin treatment in MDA-MB 231 cells (BRCA-proficient) (Figure 4C). RAD51 induction following carboplatin treatment was innately impaired in parental HCC1937 cells (BRCA1 frameshift mutant). We observed that RAD51 induction was rescued by BRCA1 wild-type expression but not by the expression of the BRCA1 L1780P mutant in HCC1937 cells (Figure 4D). We also compared RAD51 mRNA expression levels by qRT-PCR after treatment with olaparib and carboplatin. The results showed that RAD51 mRNA expression was also downregulated in BRCA1 L1780P mutant cells following carboplatin treatment (Figure 4E,F and Appendix A). These data collectively suggest that the BRCA1 L1780P mutation impairs the induction of RAD51 expression following treatment with the platinum-based drug, which is a key player in HR repair processes.

### 3.5. Nuclear RAD51 Recruitment Is Decreased in BRCA1 L1780P Mutated Cells

RAD51 translocates to DNA damage foci in the nucleus and binds to the BRCA1-PALB2-BRCA2 complex in the nucleus following DSB events. We performed IF staining of RAD51 in BRCA1-mutant cell lines. HCC1395 cells with BRCA1 and BRCA2 mutations showed significant downregulation of RAD51 levels in the nucleus following olaparib and carboplatin treatment (Figure 5A,B). The IF assay revealed that recruitment of RAD51 to the nucleus was significantly decreased in BRCA1 L1780P mutant cells following carboplatin treatment compared to that in BRCA1 wild-type cells. (Figure 5C,D and Appendix A). These data indicated that the BRCA1 L1780P mutation perturbed the nuclear localization of RAD51, in addition to decreasing total protein expression. Next, we evaluated the interaction between p-ATM and BRCA1 in BRCA1 mutant cells following carboplatin treatment. Co-IP assay showed that the molecular interaction between BRCA1 and p-ATM was decreased in BRCA1 L1780P mutant cells (Figure 5E,F). These data suggest that the BRCA1 L1780P mutation impairs the molecular interaction between p-ATM and BRCA1, resulting in significant impairment of RAD51 activity.

### 3.6. Olaparib Sensitivity Induced by Downregulation of RAD51 in BRCA-Proficient Cells

To understand the association between RAD51 expression and olaparib sensitivity, we performed RAD51 knockdown experiments. We knocked down RAD51 in the parental MDA-MB-231 (BRCA-proficient) cell line, which is resistant to olaparib treatment (Appendix A). RAD51 knockdown increased the sensitivity of MDA-MB-231 and HCC1937 cells to olaparib and carboplatin (Figure 6A,B and Appendix A). These data suggest that RAD51 depletion sensitizes HR-proficient cells to olaparib and carboplatin.

To investigate the mechanism of olaparib resistance in the context of BRCA1 L1780 mutation, olaparib-resistant cell lines were established from BRCA1 L1780P mutant cells by chronic treatment of cells with olaparib in vitro (Figure 6C). To confirm olaparib resistance, cell viability was evaluated by a cell proliferation assay following olaparib treatment. BRCA1 L1780P olaparib-resistant cell lines showed significantly lower olaparib sensitivity than parental cells (Figure 6D,E), confirming the successful establishment of olaparib-resistant cell lines from BRCA1 L1780P mutant cells.

### 3.7. Downregulation of RAD51 in Olaparib-Resistant Cell Lines Restored Olaparib Sensitivity

Olaparib-resistant cell lines showed significant upregulation of RAD51 following olaparib treatment (Figure 7A,B and Appendix A). Therefore, we knocked down RAD51 in BRCA1 L1780P olaparib-resistant cell lines to determine the association between RAD51 upregulation and olaparib resistance with respect to the BRCA1 L1780P mutation. Notably, olaparib sensitivity was robustly restored in olaparib-resistant BRCA1 L1780P mutant cells with RAD51 knockdown (Figure 7C,D). These results demonstrated that upregulation of RAD51 plays a crucial role in olaparib resistance in BRCA1 L1780P mutant cells, suggesting that RAD51 is a candidate molecular target for olaparib resistance.

## 4. Discussion

In this study, we investigated the molecular characteristics of the BRCA1 L1780P mutation and its implication in PARP inhibitor- and platinum-induced DNA damage response using TNBC cell lines with stable expression of BRCA1 constructs. Although the BRCA1 L1780P mutation only causes a point amino acid substitution in the BRCT domain rather than truncation of the entire protein, previous studies on structural and functional properties of BRCA1 L1780P mutants revealed that peptide binding and transcriptional activity were significantly decreased by this point missense mutation [24]. A previous mutagenesis study indicated that the L1780P missense mutation may cause defects in the HR function of BRCA1.

We used BL1, MSL, and unclassified TNBC cells to investigate BRCA1 L1780P mutation and these cell lines showed different biological features [30,31]. We found that BRCA1 L1780P mutant cell lines showed higher cell migration and invasion capacity than the BRCA1 wild-type cells. The different characteristics of cell lines may influence cell migration and invasion capacity. Previous studies have reported that BRCA1 plays a direct role in the proliferation and differentiation of mammary epithelial cells, in addition to HR. STAT5B and FANCA genes, which are associated with a mammary differentiation factor, were downregulated in BRCA1 depleted cells. However, proliferation markers, such as HMGA2, angiopoietin-1, and CaM kinase II, were upregulated in BRCA1 depleted cells [32,33,34]. Moreover, BRCA1 germline mutations affect tumor initiation, progression, and metastasis in breast cancer [35]. Cancer associated fibroblasts (CAFs) comprise the tumor microenvironment (TME), which is a modulator of tumor initiation, progression, and metastasis [36,37]. Previous studies have revealed that BRCA1 mutation contributes to breast cancer development by creating a pro-tumorigenic niche and affects the transformation of CAFs to metastasis associated fibroblasts (MAFs). MAFs induce metastatic changes in breast cancer cells. The effect of the BRCA1 L1780P mutation on breast cancer cell invasion and migration is consistent with that of other deleterious nonsense or frameshift mutations in BRCA1, suggesting that the BRCA1 L1780P mutation directly promotes TNBC cell invasion and metastasis.

Next, we investigated the effect of the BRCA1 L1780P mutation on the HR capacity of TNBC cells. Olaparib sensitivity was higher in BRCA1 L1780P mutant cells than in the wild-type cells, demonstrating that the BRCA1 L1780P mutation also poses synthetic lethality upon PARP inhibition [38,39]. Of note, RAD51 levels were significantly downregulated in BRCA1 L1780P mutant cells following carboplatin treatment. The translocation of RAD51 to the nucleus is a critical process in the DNA DSB repair pathway [40,41]. In addition to decreased expression, BRCA1 L1780P mutant cells showed impaired nuclear recruitment of RAD51 following carboplatin treatment compared to the wild-type cells. These results show that the BRCA1 L1780P mutation impairs RAD51 activation and nuclear translocation, which is a central node in the DNA response pathway.

Taken together, we expected that RAD51 downregulation would impair the HR signaling pathway in BRCA1 L1780P mutant cells. Notably, BRCA1 L1780P mutant cells showed decreased protein–protein interaction between p-ATM and BRCA1 following carboplatin treatment. These results suggest that the BRCA1 L1780P mutation is associated with the impairment of the molecular interaction between p-ATM and BRCA1 resulting in decreased RAD51 recruitment to the nucleus upon treatment with either olaparib or carboplatin. Consequently, the BRCA1 L1780P mutation induced HR defects in DNA DSBs (Figure 8).

Previous studies have reported that RAD51 serves as a biomarker for predicting drug sensitivity to PARP inhibitors in patient-derived xenograft models and patients [42,43]. RAD51 is closely related to HRD and drug sensitivity to PARP inhibitors, reflecting the HR status of tumor cells [44]. Our study also underscores the important role of RAD51 in BRCA1 L1780P mutation-induced HRD.

Olaparib is now a standard of care in patients with BRCA 1/2 mutation, who have progressed on previous taxane- or anthracycline-based chemotherapy [45,46]. However, resistance to olaparib frequently occurs in patients with BRCA1/2 mutation, limiting the median progression-free survival to 6–7 months [47,48,49]. To investigate the mechanism of olaparib resistance, we established olaparib-resistant cell lines from BRCA1 L1780P mutant cells. We observed upregulation of RAD51 expression in olaparib-resistant cells lines following olaparib treatment. To evaluate the relationship between RAD51 level and olaparib sensitivity, we knocked down RAD51 in BRCA wild type and mutant olaparib-resistant cells. RAD51 knockdown restored the sensitivity of olaparib-resistant cell lines to olaparib. These data demonstrate that RAD51 upregulation is a key factor for olaparib resistance in BRCA1 L1780P mutant TNBC cells.

Previous studies reported that RAD51 upregulation independently affects cell survival regardless of BRCA status following X-ray exposure and cisplatin treatment [50,51], and that olaparib resistance occurs by RAD51 upregulation [52]. These findings are consistent with the current findings.

BRCA1 L1780P mutation impaired the induction of RAD51 following DSB in TNBC cell lines. Notably, olaparib-resistant cell lines showed upregulation of RAD51 expression following olaparib treatment, and knockdown of RAD51 restored olaparib sensitivity, both in BRCA-proficient and olaparib-resistant cell lines. These data suggest that RAD51 is a crucial molecular factor in the HR pathway and is closely associated with olaparib resistance.

## 5. Conclusions

In conclusion, we explored the molecular significance of BRCA1 L1780P BRCT domain missense mutation in TNBC cells and found increased migration and invasion capacity, and HR impairment in BRCA1 L1780P mutant cells. The BRCA1 L1780P mutation caused downregulation and decreased nuclear recruitment of RAD51 following DNA damage stimulus. Protein–protein interaction of p-ATM and BRCA1 was decreased in BRCA1 L1780P mutant TNBC cells, which may lead to impaired recruitment of RAD51 in these cells. BRCA1 L1780P mutant olaparib-resistant cells showed upregulation of RAD51 expression following olaparib treatment, and knockdown of RAD51 restored olaparib sensitivity, which suggests that RAD51 is a crucial molecular factor in the HR pathway.

## Figures and Tables

**Figure 1 cancers-14-02405-f001:**
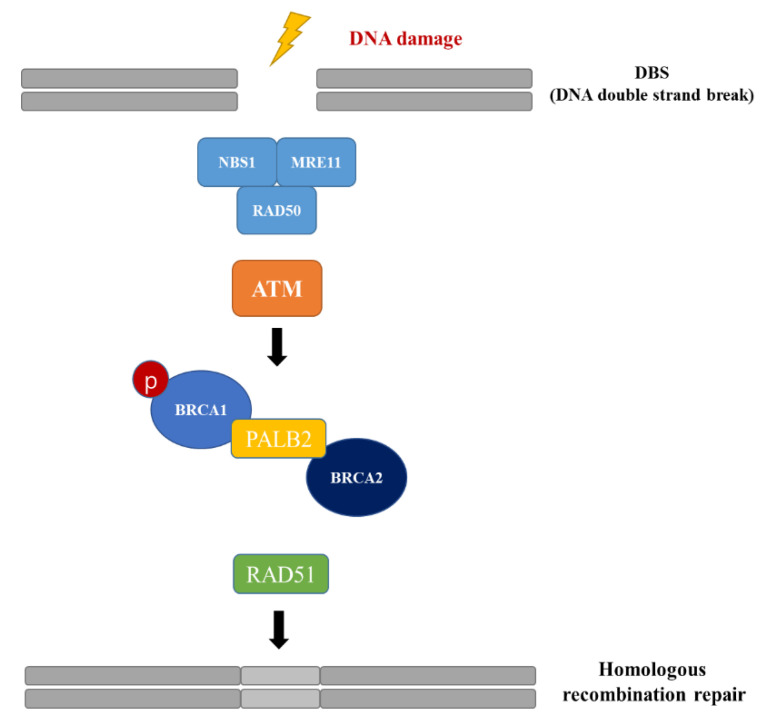
HR pathway activated by DSBs. After DSB occurs, ATM is recruited by the MRN complex (MRE11, RAD51, and NBS1) at the DSB site, and phosphorylated. A complex of BRCA1-PALB2-BRCA2 translocate to the cell nucleus and binds to RAD51.

**Figure 2 cancers-14-02405-f002:**
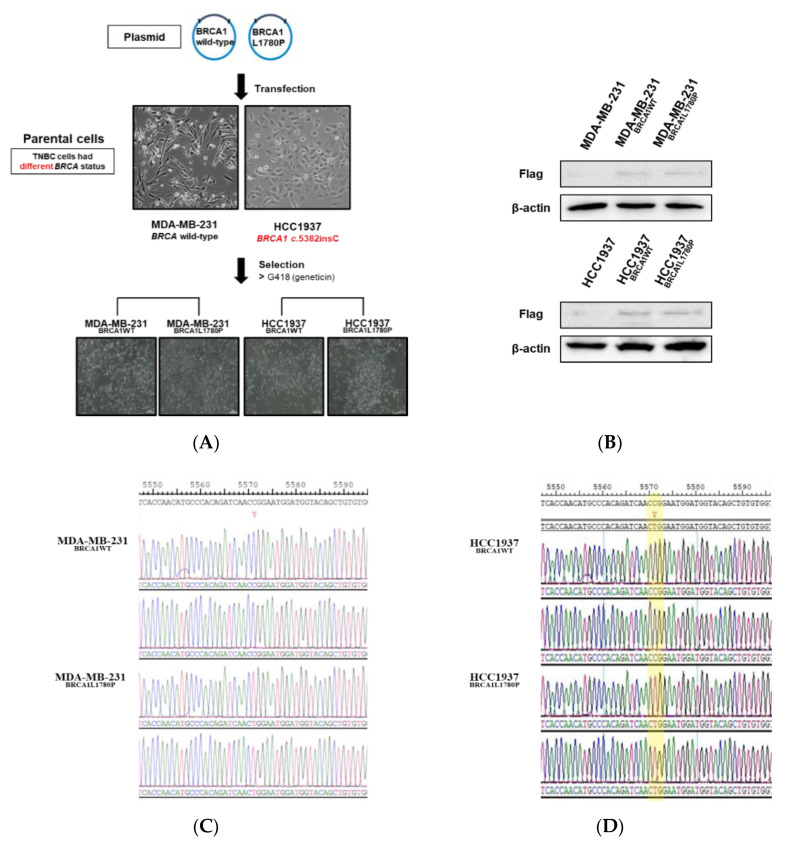
Stable expression of mutant BRCA1 proteins in TNBC cells. (**A**) Flowchart illustrating the establishment of TNBC cells with stable expression of mutant BRCA1 proteins. (**B**) Flag-tagged proteins were detected by Western blotting using anti-FLAG antibody. (**C**,**D**) Sanger sequencing was performed to confirm BRCA1 status. BRCA1 wild-type and stable BRCA1 L1780P mutant (**C**) MDA-MB-231 and (**D**) HCC1937 cells were used. (**E**,**F**) Cell proliferation assay was performed in (**E**) MDA-MB-231 parental and stably transfected cells; (**F**) HCC1937 parental and stably transfected cells were used to cell proliferation assay. Data are reported as the mean ± standard deviation (SD). Statistical analysis was performed using chi-square test. (* *p* < 0.05, ** *p* < 0.01).

**Figure 3 cancers-14-02405-f003:**
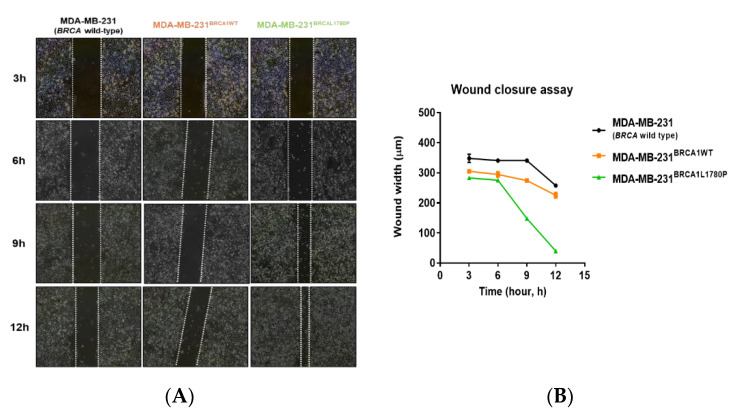
Cell migration of TNBC cell lines stably expressing BRCA1. (**A**) Cell migration assay in stable BRCA1-expressing cell lines. (**B**) Calculation of the wound closure area in stable BRCA1-expressing cell lines. Data are reported as the mean ± standard deviation (SD). (**C**,**D**) Cell invasion assay in stable BRCA1-expressing cell lines. (**C**) MDA-MB-231 parental and stably transfected cells. (**D**) HCC1937 parental and stably transfected cells. Data are reported as the mean ± standard deviation (SD). Statistical analysis was performed using the chi-square test. (* *p* < 0.05, ** *p* < 0.01).

**Figure 4 cancers-14-02405-f004:**
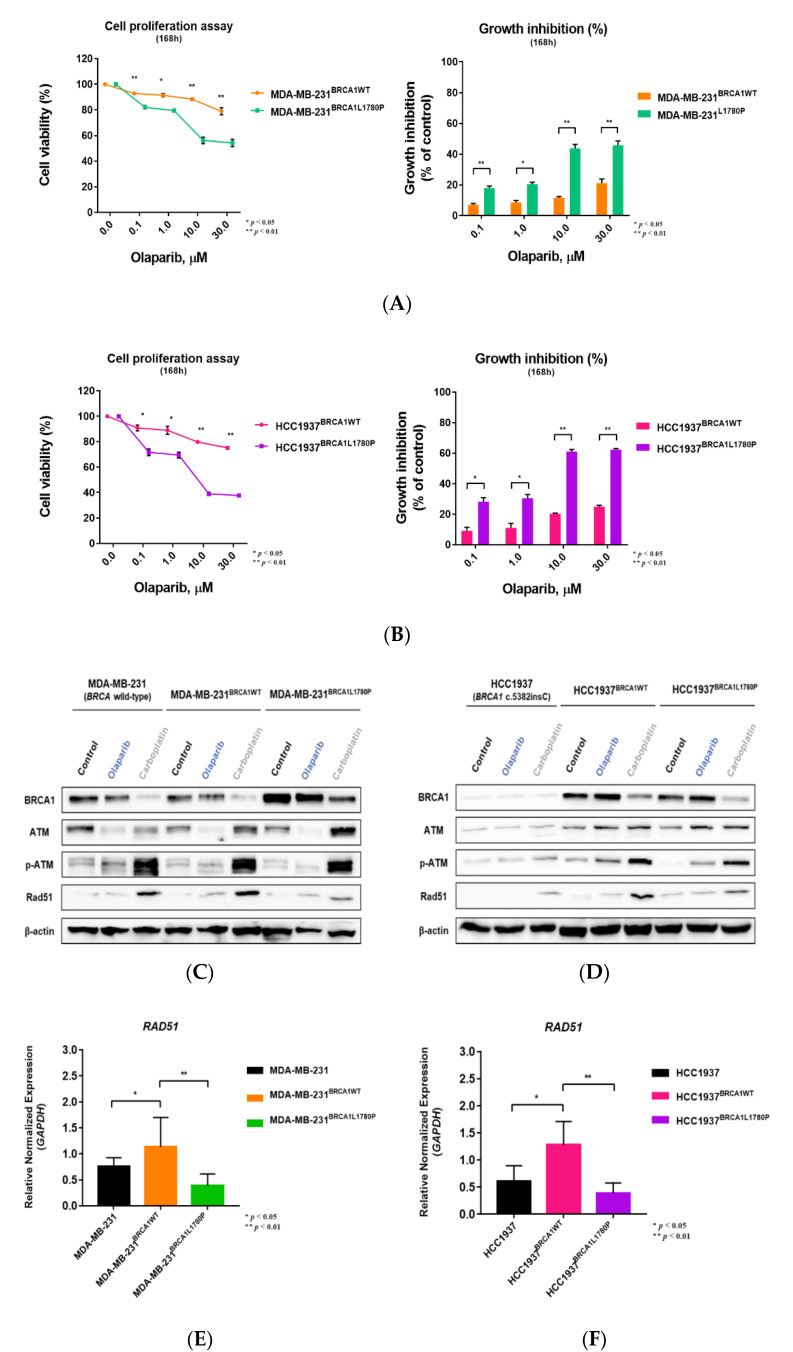
Olaparib sensitivity in stable cell lines stably expressing BRCA1. (**A**,**B**) Cell viability and growth inhibition following olaparib treatment in BRCA1-expressing cell lines. (**C**,**D**) BRCA1 L1780P mutant cells showed downregulation of RAD51 expression following carboplatin treatment. (**E**,**F**) RAD51 mRNA expression in BRCA1 L1780P mutant cells was downregulated following carboplatin treatment. Cells stably expressing BRCA1 wild-type and L1780P mutant were used. (**A**,**C**,**E**) MDA-MB-231 and (**B**,**D**,**F**) HCC1937 cells. Data are reported as the mean ± standard deviation (SD). Statistical analysis was performed using the chi-square test. (* *p* < 0.05, ** *p* < 0.01).

**Figure 5 cancers-14-02405-f005:**
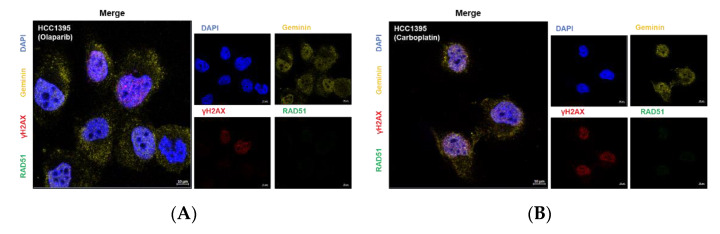
RAD51 recruitment to the nucleus and protein–protein interaction between p-ATM and BRCA1 in stable BRCA1-expressing cells following olaparib and carboplatin treatment. (**A**,**B**) RAD51 recruitment was decreased in HCC1395 cells following treatment with (**A**) olaparib and (**B**) carboplatin. (**C**,**D**): (**C**) RAD51 expression level was restored in cells expressing exogenous BRCA1 wild-type following carboplatin treatment. (**D**) BRCA1 L1780P mutant cells showing decreased RAD51 recruitment to the nucleus following carboplatin treatment. (**E**,**F**) Protein–protein interaction between p-ATM and BRCA1 was decreased in BRCA1 L1780P mutant cells following carboplatin treatment. BRCA1 wild-type and L1780P mutant were stably expressed in (**E**) MDA-MB-231 and (**F**) HCC1937 cells.

**Figure 6 cancers-14-02405-f006:**
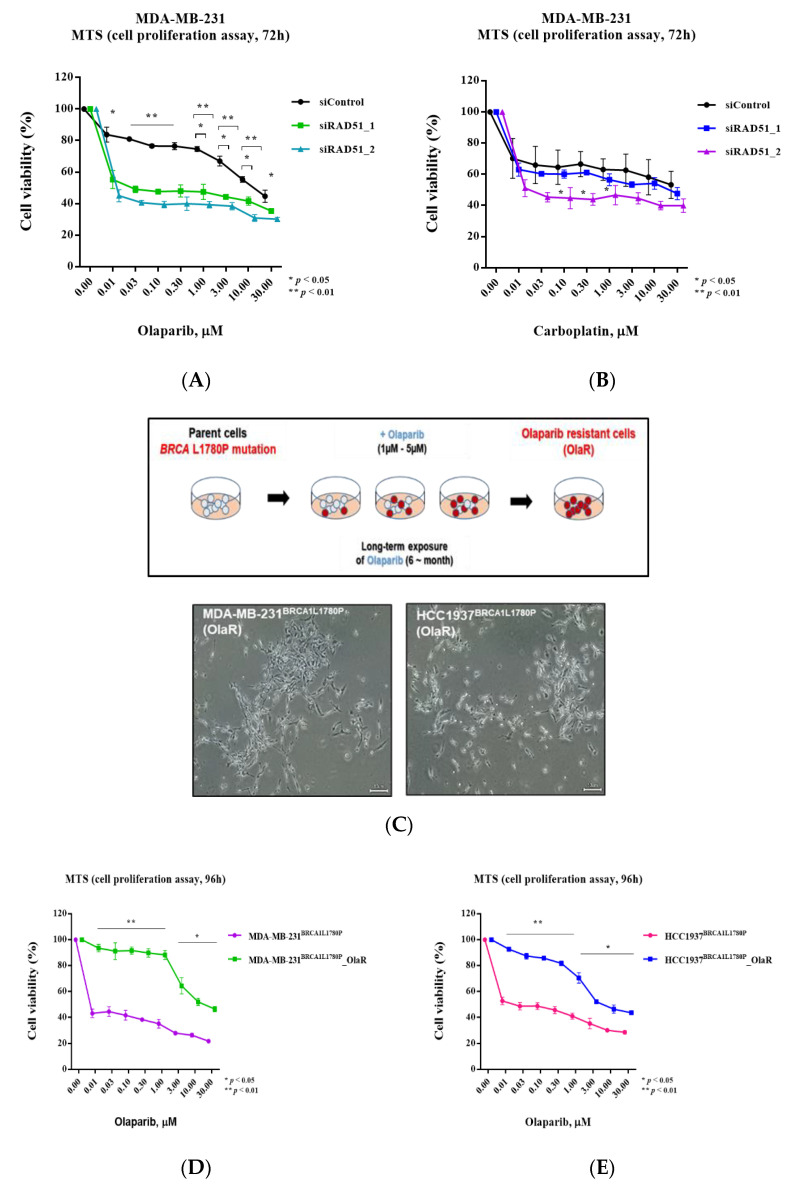
RAD51 expression level affected olaparib sensitivity in BRCA-proficient cells. (**A**,**B**) Downregulation of RAD51 expression affected sensitivity to (**A**) olaparib, and (**B**) carboplatin in MDA-MB-231 cells. (**C**) Schematic diagram showing the establishment of olaparib-resistant cell lines. (**D**,**E**) Cell proliferation assay was performed to evaluate the cell viability of parental and olaparib-resistant cell lines following olaparib treatment. Olaparib-resistant cells derived from cells stably expressing BRCA1 L1780P mutant in (**D**) MDA-MB-231 and (**E**) HCC1937 cell lines. Data are reported as the mean ± standard deviation (SD). Statistical analysis was performed using the chi-square test. (* *p* < 0.05, ** *p* < 0.01).

**Figure 7 cancers-14-02405-f007:**
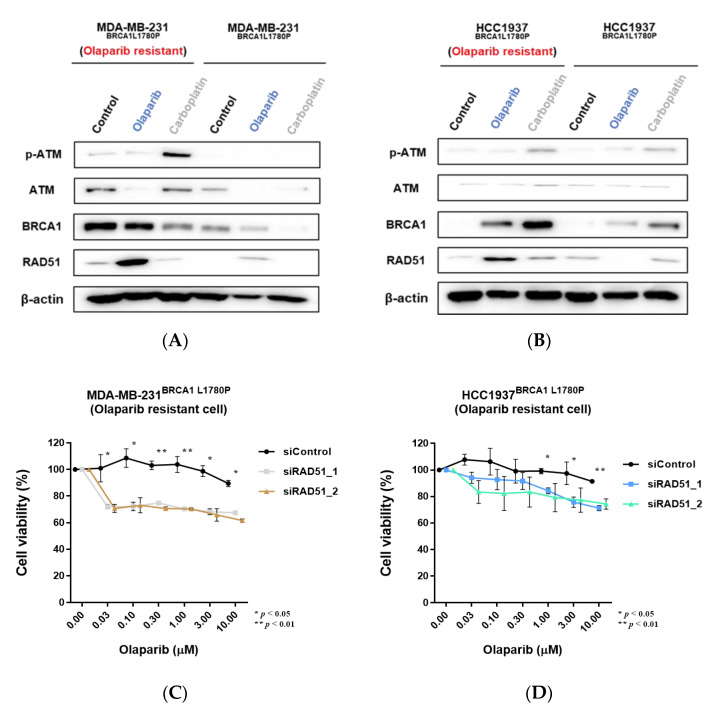
Downregulation of RAD51 expression affects olaparib sensitivity of olaparib-resistant cells. (**A**,**B**) Upregulation of RAD51 expression affects olaparib resistance in BRCA1 L1780P mutant cells derived from (**A**) MDA-MB-231 and (**B**) HCC1937 cells. (**C**,**D**) Knockdown of RAD51 in olaparib-resistant cells restored olaparib sensitivity following olaparib treatment. Olaparib-resistant cells derived from (**C**) MDA-MB-231, and (**D**) HCC1937 cells. Data are reported as the mean ± standard deviation (SD). Statistical analysis was performed using the chi-square test. (* *p* < 0.05, ** *p* < 0.01).

**Figure 8 cancers-14-02405-f008:**
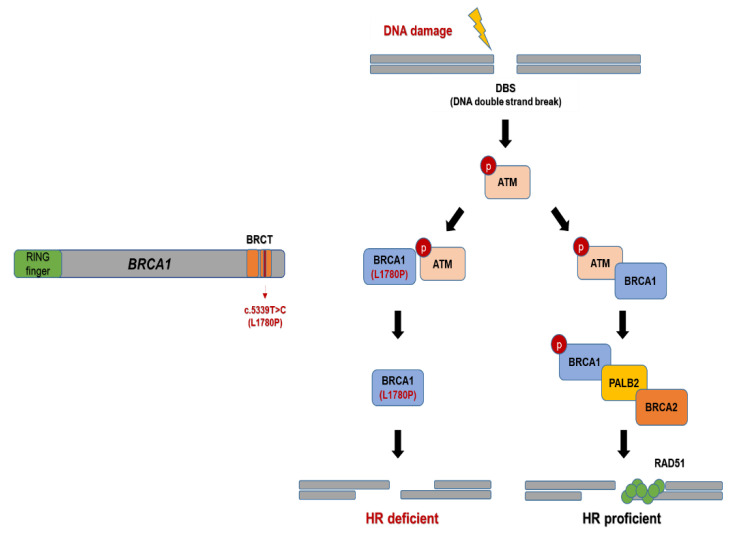
HR pathway in BRCA1 L1780P mutated TNBC cells. When DSB occurs, BRCA1 L1780P mutation affects the binding affinity between p-ATM and BRCA1. Therefore, RAD51 recruitment is decreased in the cell nucleus. Consequently, HRD is induced in BRCA1 L1780P mutated cells.

## Data Availability

The data in this article are available.

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
