# Peer review of "Molecular Characterization of BRCA1 c.5339T>C Missense Mutation in DNA Damage Response of Triple-Negative Breast Cancer"

_cancers, 2022, doi:10.3390/cancers14102405_

Round 1

Reviewer 1 Report

Lee JD et al. conducted a cell biological study of BRCA1 c.5339T>C missense mutation (L1780P) in two triple-negative breast cancer (TNBC) cell lines.  The authors suggested that BRCA1 L1780P mutant had effects on cell proliferation, motility, invasiveness, and sensitivity to olaparib and carboplatin.  In addition, they examined molecular and cellular alterations by olaparib and carboplatin in these mutant cell lines.  Although the contents of the manuscript appear to present convincing data on the biological meaningfulness of BRCA1 L1780P mutant in TNBC cell lines.  However, there are still several unclear point to support their conclusions.

  1. In Abstract, they mention about HCC1395 TNBC cell lines, but in text, HCC1395 cell line was only used for the immunofluorescence assay shown in Figures 4A to 4D.  Because there are no other experimental data of HCC1395 BRCA1 L1780P mutant throughout the manuscript, it appears unnatural to use the data of HCC1395 only there.  In HCC1395 cell line, not only BRCA1 but also BRCA2 are impaired, and the effect of aberrant BRCA2 might be present even after transfection of BRCA1 L1780P mutation.  Therefore, they should present immunofluorescence data of HCC1937 and/or MDA-MB231 with BRCA1 L1780P mutations as these figures.
  2. From Figure 4, they should present immunofluorescence images of RAD51 and γH2AX localization not only for the status after treatment with olaparib or carboplatin but also for the status before the treatment for comparison. If it were not for the images of pretreatment status, it is difficult to mention that BRCA1 L1780P mutant cells showed impaired nuclear recruitment of RAD51 following carboplatin treatment compared to the wild-type cells as described in Discussion, line 373-374.
  3. In Figure 3E, they should also describe whether or not RAD51 expression level was significantly different between MDA-MB-231 BRCA1WT and MDA-MB-231 BRCA1L1780P. Likewise, in Figure 3F, they should also describe whether or not RAD51 expression level was significantly different between HCC1937 and HCC1937 BRCA1WT. 
  4. In Figure 3, data of olaparib (Figures 3A, 3B, 3C, and 3D) and data of carboplatin (3C-3F) are mixed together but appear mutually incomplete. It is unclear why there are no data on carboplatin in Figures 3A and 3B, whereas why there are no data on olaparib in Figures 3E and 3F.
  5. In Figure 5A, 5B, 5D, 5E, 6C, and 6D, the authors should describe if there were statistical differences between curves with providing p values.
  6. In Figure 6, the authors present the data of changes in cellular viability by the introduction of siRAD51 in MDA-MB231 BRCA1L1780P and HCC1837 BRCA1L1780P only. However, in order to know the effect of RAD51 knock-down in these mutant cells, they should compare these results with the effect of RAD51 knock-down in MDA-MB231 or MDA-MB231 BRCA1WT and in HCC1837 or HCC1837 BRCA1WT.
  7. In Figure 2C, “Cell invastion assay” should be “Cell invasion assay”

Reviewer 2 Report

The manuscript entitled “Molecular Characterization of BRCA1 c.5339T>C Missense Mutation in DNA Damage Response of Triple-Negative Breast Cancer” by Jeong Dong Lee et al., could be an interesting work since as the authors point “aims to reveal the molecular mechanisms and therapeutic relevance of BRCA1 L1780P mutation in DNA damaging response of triple-negative breast cancer.” However, in my opinion, there are a few major issues the authors should address before acceptance for publication.

The authors state in the abstract and I quote “We established MDA-MB 231, HCC1937, and HCC1395 TNBC cell lines expressing BRCA1 L1780P mutant.” After this statement, the authors show their results comparing BRCA1 L1780P mutant TNBC cells vs the BRCA1 wild-type cells. However, although the results may look quite convincing, they seem to be incomplete. Three cell lines are a limited number to prove their statements, although it could work if experiments are reproduced and validated in all of them, the authors only show the results in some of the cell lines while there are no data in others. The authors mainly showed the results in two cell lines the MDA-MB 231 and HCC1937 whereas no data on HCC1395 are shown. Even more, when results on MDA-MB 231 and Hcc1937 are shown, sometimes they use three conditions (Parental cell line and stably-transfected cells BRCA1WT and BRCA1 L1780P) whereas sometimes they don’t show results on parental cells and only results on stably-transfected ones are shown:

1.- In figure 1 only results on MDA-MB 231 and HCC1937 are shown, there is no data or results on HCC1395.

2.- Authors state that “BRCA1 L1780P mutant cells showed low cell proliferation capacity compared to BRCA1 wild-type MDA-MB-231 and HCC1937 cells”. In Figure 1 E, is shown that there are no differences in stable-transfected cells whereas the significant differences are with the parental cell line. In this case, the low proliferation rate could be due to the transfection rather than the mutation status of the BRCA1, otherwise, BRCA1WT proliferation rates should be closer to the parental BRCA1 wildtype and different from BRCA1 L1780P. Could you explain this better?

3.- The results showing differences in proliferation rates after Olaparib treatment in Figures 3A and 3B are only the stable-transfected whereas no data on parental cells are shown

4.- The IF assay in Figure 4A-D is done only in cell line HCC1395 whereas this cell line is not used in any other experiment or result shown in the draft. In fact, Figure 4E-F are again the results of HCC1937 and MDA-MB231. Why is that so? Authors should show the results in all the cell lines with all conditions.

5.- Another point to be considered and something that authors should discuss is the nature of the different TNBC cell lines used and how these differences may influence their results. If we take into account that they used mainly HCC1937 and MDA-MB231, they should compare the differences of these cells in the discussion. For example, there are many works discussing the characteristics of the different cell lines since HCC1937 is a Basal-like type 1 and MDA-MB231 is a more mesenchymal type according to Lehmann classification among others. The authors should consider the use of these papers in the discussion and/or introduction of the manuscript.

(Lehmann et al 2011 PMID: 21633166 PMCID: PMC3127435 DOI: 10.1172/JCI45014)

(Dai et al 2017 J Cancer. 2017 doi: 10.7150/jca.18457 PMID: 29158785)

(Smith SE et al 2017 PMID: 28583138 PMCID: PMC5460504 DOI: 10.1186/s13058-017-0855-0)

6.- In the same direction, since authors mainly use these two cell lines, they should consider also discussing the differences between HCC1937 and MDA-MB231 regarding p53 status. This is important because p53 has a very important role in HR DNA damage and RAD51 regulation at the transcription and functional levels. They should consider it because HCC1937 has a WT status for p53 and low protein expression whereas MDA-MB231 has p53 mutated (R280K) and protein overexpression.  In fact, Orhan E et al, published in 2021 in Cancers ((Cancers 2021, 13, 2930. https://doi.org/10.3390/ cancers13122930) nice work on the regulation on RAD51 on the transcriptional level and they state that “Other identified RAD51 transcription regulators comprise EGR1, as an activator, and p53 as a repressor. Interestingly, while the normal function of phosphorylated p53 is to repress RAD51 expression, the p53 contact mutant R280K loses this ability and does not reduce DNA-damage-induced RAD51 foci, possibly explaining the resistance to chemotherapy of p53-mutated cancers.

To cite and use this work among others related to the p53 status in the cell lines would improve the discussion and should be considered.

MINOR COMMENTS.

As for minor considerations, authors should check the draft carefully since they have two figures 1, one in the introduction (lines 70-75) and another in the results (lines 227-228)  in the paper, and also they cite a Table1A in line 76 that I could not find anywhere.

Reviewer 3 Report

In this manuscript, the authors have investigated the role of BRCA1 L1780P mutation in TNBC cell lines (mainly in MDA-MB-231 and HCC1937 cells). Their results indicated that the mutation impairs Rad51 activity by disrupting p-ATM and BRCA1 interaction, suggesting the possible target of Rad51 in BRCA1 L1780P TNBC. However, there are some remaining questions to be answered:

1, The authors transfected BRCA1 wt and BRCA1 L1780P in multiple TNBC cell lines although these cells have different BRCA1 backgrounds (MDAMB231: BRCA1 wt, HCC1937, BRCA1 c.5382 insC). Does different BRCA1 backgrounds affect the result?

  • For example, in Fig 1E-F, cDNA transfection of wt and L1780P plasmids both inhibit the cell proliferation in MDAMB231 while transfection of wt plasmids only showed less inhibition in HCC1937 instead.
  • In Fig 3C-D, two cell lines have different basal levels of BRCA1, ATM, Rad51, p-ATM.
  • In Fig 5A-B, knocking down Rad51 have different impact on cell viability (more obvious in MDAMB231).

2,  in Fig 3, the authors decided to examine the cell proliferation by comparing transfected cells. Could the authors explain why they don’t include parental cells?

3, For western blot, could the authors also quantify the band intensity? For example, in Fig 3F, the mRNA level of Rad51 in L1780P cells decreased compared with wt or parental cells after drug treatment, However, in Fig 3D, it seems like the protein level of Rad51 in L1780P cells actually increased compared with parental cells with or without drug treatment (Olaparib or carboplatin). Does mutation increase basal Rad51 level?

4, in Fig 4, the authors claimed that Rad51 recruitment is decreased in L1780P cells. Does the decreased recruitment of Rad51 induced by the decreased Rad51 expression (Fig 3)? Or it is indeed impair the translocation? Have the authors performed the cellular fractionation experiments to measure the Rad51 expression in different cellular components?

5, Some figures are missing p-values.

6, The authors suggested that Rad51 downregulation restore sensitivity to Olaparib in Olaparib-resistant L1780P cells (Fig 6). Have the authors examined the effect of upregulation of Rad51 in parental cells or L1780P cells? Are cells more resistant to Olaparib?

7, In Fig 7, it seems like L1780P mutation caused the less interaction with p-ATM, then decreased p-BRCA1, less interaction of PALB2 and BRCA2. Has the authors examined the phospho-BRCA1 level and association with downstream partners( PALB2 and BRCA2) after carboplatin treatment?

8, Some typo errors. For example,

  • Line 76, Table A1.? I cannot find the information of Table A1.
  • Line 106, missing “10%” after “FBS +”
  • Line 120, missing “cells/well”
  • Line 122, use “h”, instead of “hr”
  • Line 112, 128, 172, “105”, not “105”,
  • Line 134, “103”, not “103”
  • Line 182 and 197, “104”, not “104”
  • Line 191, Table S1, not Table 1

Round 2

Reviewer 1 Report

The questions and comments have been answered appropriately.

Reviewer 2 Report

It is clear to me that the authors answered the best they could the comments I asked for, however they stated in two of the requested points, that they will prove them in further studies. For instance, the first part about proliferation is not clear to me, there are no differences and the authors just answered and I quote " In these data, it was not enough to explain the cell proliferation capacity according to different BRCA1 status. In a further study, we will investigate the relationship between cell proliferation capacity and different BRCA1 status".  In the same line of work, they don't prove the relevance of a 3rd cell line the HCC1395 and that's why all the experiments on this cell line are mainly in supplementary data.  However, they proved and changed all the other points requested. At this point, I decided to accept it in its present form.